# Receipt of a combined economic and peer support intervention and clinical outcomes among HIV-positive youth in rural Rwanda: A retrospective cohort

**Dale A. Barnhart**[1,2]*, **Josée Uwamariya**[1], **Jean Népomuscène Nshimyumuremyi**[1], **Gerardine Mukesharurema**[1], **Todd Anderson**[1], **Jean d'Amour Ndahimana**[1], **Vincent K. Cubaka**[1], **Bethany Hedt-Gauthier**[2,3]

1 Partners In Health, Rwinkwavu, Rwanda, 2 Global Health and Social Medicine, Harvard Medical School, Boston, MA, United States of America, 3 Biostatistics, Harvard T.H. Chan School of Public Health, Boston, MA, United States of America

* dale_barnhart@hms.harvard.edu

## Abstract

### Introduction

To combat poor clinical outcomes among HIV-positive youth, Partners In Health/Inshuti Mu Buzima (PIH/IMB) implemented Adolescent Support Groups (ASGs), which combined peer support and group-based economic incentives to promote treatment adherence, economic empowerment, and viral suppression. This study assesses the association between ASG membership and clinical outcomes among HIV-positive youth living in rural Rwanda.

### Methods

We constructed a retrospective cohort using PIH/IMB's electronic medical record (EMR) system. ASG members were matched to control youth within strata defined by health facility, year of birth, and whether the patient had enrolled in HIV services as a pediatric patient, as a PMTCT mother, or through another route. Our 12-month outcomes of interest were a) death-free retention in care, b) death-free retention with active follow-up, c) ≥80% adherence to appointment keeping, and d) viral load suppression (<20 copies/ml). We used generalized linear mixed models to estimate odds ratios for the association between ASG participation and each outcome. To mitigate possible unmeasured confounding, we additionally included participant data from the previous year and conducted a difference-in-difference analysis for each outcome to assess whether ASG members experienced greater changes compared to control youth over a similar period.

### Results

Two-hundred sixty ASG members were identified in the EMR and matched to 209 control youth for analysis. After 12 months of follow-up, ASG members had similar outcomes to the control youth in terms of death-free retention (93% vs. 94%), death-free retention with active

**Data Availability Statement:** This analysis using routinely collected electronic medical record data

and contains sensitive information about a vulnerable patient population. Our ethical approvals from the Rwanda National Ethics Committee require this data to be stored on a secure PIH/IMB server located within Rwanda or on password-protected computers of study collaborators. Researchers with a reasonable request for the data presented in this study should submit their request to Inshuti Mu Buzima Research Committee (IMBRC, imbrc@pih.org) to identify additional steps for approval.

**Funding:** The ASG program was funded by Partners in Health/Inshuti Mu Buzima. DAB is supported by the Harvard Medical School Global Health Equity Research Fellowship, funded by Jonathan M. Goldstein and Kaia Miller Goldstein. This publication was made possible with help from the Harvard University Center for AIDS Research (CFAR), an NIH funded program (P30 AI060354), which is supported by the following NIH Co-Funding and Participating Institutes and Centers: NIAID, NCI, NICHD, NIDCR, NHLBI, NIDA, NIMH, NIA, NIDDK, NINR, NIMHD, FIC, and OAR. The content is solely the responsibility of the authors and does not necessarily represent the official views of the National Institutes of Health. The funders had no role in study design, data collection and analysis, decision to publish, or preparation of the manuscript.

**Competing interests:** I have read the journal's policy and the authors of this manuscript have the following competing interests: DAB, JU, JNN, GM, TA, JAN and VKC are all employed by Partners In Health, which is the NGO that implements the Adolescent Support Group program. There are no other conflicts to declare.

follow-up (79% vs. 78%), ≥80% adherence to appointment keeping (42% vs. 43%), and viral suppression (48% vs. 51%). We did not observe any significant associations between ASG participation and clinical outcomes in crude or adjusted models, nor did ASG members experience greater improvements than control youth in our difference-in-difference analysis.

## Conclusions

The ASG program did not improve retention, appointment adherence, or viral suppression among HIV positive youth in rural Rwanda. Challenges implementing the intervention as designed underscore the importance of incorporating implementation strategies and youth perspectives in program design. This population remains vulnerable to poor clinical outcomes, and additional research is needed to better serve youth living with HIV.

## Introduction

Over 25% of new HIV infections occur among young people between 15 and 24 years of age and record numbers of peri/postnatally infected children are surviving to adolescence [1, 2]. Providing high-quality care to youth living with HIV is critical for achieving the 90-90-90 goals and ending the AIDS epidemic [3, 4]. However, compared to other age groups, youth living with HIV are less likely to be retained in care [5, 6], less adherent to ART [7], experience worse virologic outcomes [8, 9], and face elevated risk of mortality [10, 11].

Relatively few interventions designed to improve adherence among youth with HIV have been rigorously studied. Peer-support has been consistently identified by adolescents and their clinicians as a key factor in promoting treatment adherence [12–14]. Although there is some evidence that peer support programs can improve clinical outcomes among youth with HIV, substantial heterogeneity between programs and infrequent evaluations of existing peer support groups remain a substantial barrier to understanding and improving the implementation of these interventions [15]. Similarly, although economic challenges are often cited as a barrier to treatment adherence, very few youth-focused intervention have assessed the impact of providing economic support on treatment outcomes [16–19]. Furthermore, although interventions designed to promote treatment adherence and clinical outcomes among youth are commonly include multiple complementary components [20], only two previous studies have combined peer support and economic incentives to support youth with HIV, and both were small pilot studies designed to assess feasibility and safety rather than effectiveness [21, 22].

In Rwanda, virologic failure among youth with HIV is six times higher than in adults [23]. Only 58% of youth in rural districts have good appointment keeping and only 13% reported optimal adherence to ART [24]. To combat poor adherence and poor clinical outcomes among youth, Partners In Health/Inshuti Mu Buzima (PIH/IMB) implemented Adolescent Support Groups (ASGs), a complex intervention designed to combine peer support groups and group-based economic incentives to promote treatment adherence, economic empowerment and viral suppression among youth aged 15–24 years living with HIV. This retrospective cohort study assesses the ASG program's effectiveness by comparing 12-month clinical outcomes among ASG members to other HIV-positive youth living in rural Rwanda.

## Methods

### Study setting

This study was implemented in the catchment areas of the Butaro, Kirehe, and Rwinkwavu District Hospitals, which are located in Burera, Kirehe, and Kayonza districts, respectively. These three public hospitals and their affiliated primary-level health centers are located in rural Rwanda and receive health system strengthening support from PIH/IMB. The ASG program was designed and implemented by PIH/IMB in partnership with the clinical staff at participating health facilities.

### The Adolescent Support Group program

The ASG program was first launched in Kayonza district in July 2017 and was later expanded to Burera and Kirehe districts in July 2018. Originally, the program was designed so that HIV-program nurses at each participating health facility were asked to refer economically vulnerable youth between the 15–24 years to the ASG program. At the start of the program, each health facility founded a single ASG. Groups were allowed to be mixed in terms of both sex and age, but were expected to include 10–15 members and meet once per month. Although clinicians were instructed to prioritize orphans and out-of-school youth, referrals were ultimately based on clinician judgment and strict enrollment criteria were not enforced. While not all youth were invited to participate, records on who clinicians chose to invite and whether any invited youth declined to participate are not available. Imperfect implementation led over 20% of the youth falling outside the target age range of 15 to 25 years of age and group sizes ranging from 2 to 33. Group composition was also very heterogeneous, with mean age of group members ranging from 11 to 25 and the proportion of male members ranging from 0–67%.

During group meetings, peer support was provided through nurse-facilitated group discussions related to ART adherence as well as other social and economic challenges. Economic incentives were deposited into groups' joint savings accounts based on group achievement on three key indicators: a) pharmacy attendance; b) achieving a pre-defined bi-annual savings targets; c) and annual viral suppression, which was initially defined as <20 copies/ml but later relaxed to <200 copies/ml in line with national guidelines. For each indicator, funding awarded decreased as group performance dropped in a stepwise fashion that was designed to encourage positive peer pressure among group members (Table 1). After two years of saving, the groups were mentored on using their savings to start an income-generating activity, such as livestock farming. In practice, provision of incentives went through planned and unplanned adaptations. However, key deviations included sporadic doubling of incentives and providing 15,000 RWF for each individual virally suppressed patient or patient designated "stable" in the national Differentiated Service Delivery Model rather than according to the stepwise group-based incentive scheme. Additional details on the adaptation of the intervention, group characteristics, group composition, group achievement on indicators, and group earnings have been reported elsewhere [25].

### Data sources

We constructed a retrospective cohort using existing data from PIH/IMB's electronic medical record (EMR) system, which has been used to support clinical HIV activities since 2005. ASG programmatic records were used to generate a list of beneficiaries, and which we fuzzy matched to identify ASG participants in the EMR based on name; patient identifier, either the government HIV program TRACnet number or the health facility EMR patient identifier; year of birth; and health facility. ASG programmatic records were also used to identify the date

**Table 1. Incentive structure for the Adolescent Support Group program.** All incentives were to be deposited in a group savings account to fund subsequent income-generating activities.

| Intervention design | Pharmacy attendance | Savings | Viral suppression | Stepwise incentive structure | |
|---|---|---|---|---|---|
| | | | | Group achievement on indicator | Proportion of maximum incentives earned |
| | % of scheduled visits that were attended by group members | % of savings target achieved | % of members virally suppressed | | |
| | 10,000 RWF[1] per member | 100% matched savings | 15,000 RWF[2] per member | 100% | 100% |
| | | | | 90% to <100% | 80% |
| | Awarded quarterly | Awarded bi-annually | Awarded annually | 80% to <90% | 60% |
| | | | | 70% to <80% | 40% |
| | | | | 60% to <70% | 20% |
| | | | | <60% | 0% |
| **Implementation challenges** | Implemented as planned | Implemented as planned | Viral suppression initially defined as <20 copies/ml but later relaxed to <200 copies/ml in line with national guidelines. | Incentives were sporadically doubled when money was available in the budget. | |
| | | | Missing or outdated viral load data made it difficult to calculate a group-based indicator. Some groups resorted to reporting the % of patients considered "stable" | Stepwise incentive structure often abandoned for the viral load indicator–instead, 15,000 was deposited per suppressed or stable youth. | |

[1]Approximately 10 USD

[2]Approximatly 15 USD

when beneficiaries enrolled in the program. Because the ASG intervention was rolled out in different facilities at different times, we identified specific enrollment periods for each facility. At each participating ASG facility, the enrollment period ranging from between one year before the first recorded youth enrolled into the ASG program to one year after the last recorded youth enrolled. In several facilities, all youth enrolled in the same date such that this enrollment period spanned two years; however, other groups embraced a more open enrollment strategy. Patients were considered to belong to the most frequently attended ASG-participating location they visited during that enrollment period. Non-ASG members were also assigned a "pseudostart" date selected at random from the distribution program start dates among ASG participants at their health facility. For each individual, baseline was considered to start on the patient's closest observed visit date within 90 days of their enrollment date (for ASG members) or pseudostart date (for non-members).

## Eligibility

Our analytic population consisted of individuals in the EMR database who were a) HIV-positive; b) born between 1987 and 2012; c) had not exited the HIV program through death, loss to follow-up, or external transfer prior to their enrollment date (for ASG members) or pseudostart date (for non-members); d) had attended their assigned ASG location (for ASG-members) or a ASG participating facility (for non-members) during the enrollment period; e) had at least one visit recorded in the EMR within 90 days of their enrollment date (for ASG members) or pseudostart date (for non-members); and f) did not transfer to another facility during the 12 months follow-up.

After applying this inclusion criteria, preliminary analyses revealed that ASG members were much more likely to be male, to be pediatric patients, and to be between the ages of 15–25 at enrollment than control youth, which included a large proportion of young adult women entering care through the PMTCT program. To improve the comparability of our ASG

members and control group, we matched control youth to ASG member within strata defined by health facility, year of birth (categorized as 1987–1992; 1993–1997; 1998–2002, and 2003–2012), and whether or not the patient had initially enrolled the HIV program as a pediatric patient, as a mother in the PMTCT program, or through another route. In our primary analysis, we randomly selected up to two control youth per ASG youth from each stratum. We did not explicitly match on calendar time; however, because a) non-ASG members were assigned a pseudostart date selected at random from the distribution program start dates among ASG participants at their health facility and b) only individuals with an observed visit date within 90 days of this pseudostart date were eligible to be controls, the distribution of start date for ASG members and eligible controls were constrained to be similar. Because not all strata had two control youth for every ASG youth, the distribution of age, geographic location, and type of HIV patient in the comparison population still did not perfectly reflect the ASG population after matching; however, the comparability was improved. As a sensitivity analysis, we repeated our analysis taking up to one and up to three control youths per ASG member in each stratum.

## Statistical analysis

We used frequencies and percentages to assess confounding between our ASG members and control youth. Key demographic and clinical characteristics included district; sex; age at the start of the ASG program, categorized as <15, 15–19, 20–25, and >25; patient type at enrollment in the HIV program categorized as pediatric patient, PMTCT mother, or other; distance to health center, categorized as <2 km, 2–5 km, ≥5 km, or missing; duration in HIV program categorized as <1 year, 1–5 year, or ≥5 years; ART treatment category defined as not initiated, first-line regimen, second-line regimen, or unknown regimen; last viral load results, categorized as <20, 20-<1000, ≥1000, or no prior viral load test recorded; and BMI, categorized as underweight (<18.5 kg/m$^2$), normal weight (18.5-<25 kg/m$^2$), overweight (>25 kg/m$^2$,), and missing. Differences between the two groups were assessed with Chi-square tests or, for variables with cell counts under 5, Fisher's exact tests.

Our primary outcomes of interest were a) death-free retention in care at 12 months, b) death-free retention with active follow-up at 12 months, c) ≥80% adherence to appointment keeping over a 12-month period, and d) viral load suppression at 12 months (<20 copies/ml). For all twelve month outcomes, baseline was considered to occur at the closes visit date within 90 days of their-enrollment date (for ASG members) or at the pseudostart date (for controls). Patients were considered to be retained if they had at least one observed clinical visit in the EMR after the twelve-month follow up, if ≤90 days had passed since their final appointment, or, if a final appointment had not been scheduled after their last observed visit, if ≤210 days had passed from their final observed visit. Patients were under active follow-up at the end of the study period if they were observed <90 days from their last appointment scheduled during the study period or, if a final appointment had not been scheduled after their last observed visit during the study period, <210 days from last observed visit occurring during the study period. Patients were considered to have attended a scheduled appointment if they had a recorded visit date within 7 days of the scheduled visit's date. Viral load results were included if they occurred within 6–18 months of the enrollment date. Patients who died or were not retained in care under active follow up were assumed to be not virally suppressed. Youth who were under active follow up but who did not have viral load results taken during the 6–18 month window were excluded from the complete case analyses.

We reported the number and frequencies of ASG members and control youth achieving each of the primary outcomes. Using generalized mixed models with a logit link and a random

intercept for health facility, we estimated odds ratios assessing the association between ASG participation and each outcome. We considered three sets of models: a crude analysis that included no confounders, a minimally adjusted analysis that included all demographic and clinical characteristics associated with ASG membership with a p<0.20, and a fully adjusted analysis that considered all demographic and clinical characteristics as potential confounders. Due to substantial missingness in our viral suppression outcome, we conducted both a complete case and a multiply imputed analysis, which used data on demographic and clinical characteristics to generate 25 imputations. Imputation was only conducted for the viral suppression outcome. Patients with missing data on covariates were retained in the models using a missing indicator method. In our models, we dichotomized duration in HIV program (<5 years versus ≥5 years) and ART treatment category (second line versus not second line) due to positivity issues. Because adolescents were referred to ASGs based on perceived economic vulnerability, unmeasured confounding by socioeconomic status was considered to be a possible source of bias. To mitigate this bias, we additionally conducted a difference-in-difference analysis, which is common method for analyzing health programs when randomization is not possible [26]. This analysis was used to compare changes in outcomes among ASG members to changes in outcomes among control youth in the year before and year after the intervention. To be eligible for this analysis, youth had to have enrolled in HIV care more than 12 months before the start of follow up and have an observed visiting within 90 days of that date. Our difference-in-difference models included fixed effects for being an ASG member versus a control youth, for pre- versus post-intervention status, and their interaction as well as random intercepts for youth and health facility. The interaction term was used to assess the extent to which changes observed among the ASG members after enrollment exceeded changes observed among the control youth over a similar period. We reported crude and minimally adjusted analyses for death-free retention with active follow-up, ≥80% adherence to appointment keeping, and viral load suppression and included a complete case and multiply imputed models for viral suppression. Death-free retention in care at 12 months could not be assessed in the difference-in-difference analysis because only patients who were not retained in the pre-intervention period would not have been eligible for inclusion in this study.

## Ethical considerations

This study received approval from Inshuti Mu Buzima Research Committee (IMBRC), Rwanda National Ethics Committee (NO. 150/RNEC/2020) and Harvard University approvals (IRB20-0565). Since this study relied on retrospective data sources, the requirements for written informed consent were waived.

## Inclusivity in global research

Information regarding the ethical, cultural, and scientific considerations specific to inclusivity in global research is included in the Supporting Information (S1 Checklist)".

## Results

Out of the 324 identified ASG beneficiaries, 295 (91%) were fuzzy matched to the EMR database. Of those who could not be linked, 11 (38%) were from a single facility which, for historic reasons, does not consistently use the same EMR system. Of the linked patients, 3 were later confirmed to by HIV-negative. An additionally 17 patients were excluded for reasons that may reflect incorrect fuzzy matching: 4 were born before 1987 or after 2021, 7 exited the EMR before enrollment in the ASG, 6 did not have recorded visits at their assigned ASG facility during the enrollment period. Finally, 11 did not have any clinical visits recorded within 90 days

of their enrollment date and 4 transferred to an external facility prior to the end of the 12 month follow up. The final sample size of ASG members was 260. After applying similar criteria to non-members and matching eligible control youth to the remaining ASG-members, 209 control youth were included in the study for a total sample size of 469 (Fig 1). The median start dates within this cohort was June 2018 for ASG members (range: May 2017-Sept 2019) and for control youth (range: April 2017-June 2019).

When comparing the ASG members to the control youth, ASG members were less likely to be under 15 (15.0% vs. 24.9%) or over 25 (4.6% vs. 11.0%). However, almost a fifth of ASG members fell outside the target age range for the intervention. ASG members were also more likely to be male (44.2% vs. 35.4%), to live closer to the health facility, and had been in the HIV program for a longer duration (Table 2). After 12 months of follow-up, ASG members had similar outcomes to the control youth in terms of death-free retention (93% vs. 94%), death-free retention with active follow-up (79% vs. 78%), ≥80% adherence to appointment keeping (42% vs. 43%), and viral suppression (48% vs. 51%) (Table 3). Although it was not a primary outcome, the proportion of youth with viral load <1000 copies/ml was also similar between the two groups (62.8% vs. 61.3%).

We did not observe any significant associations between ASG participation and any of our outcomes in our crude, minimally adjusted, or fully adjusted models nor did we observe any association between ASG participation and viral load suppression in our multiply imputed models (Table 4). Our difference-in-difference analysis included 413 youth who had available EMR data from the year prior to the start of the intervention. For death-free retention and viral suppression, there was no significant difference in the pre-post intervention change among ASG members compared to among control youth (Table 5). For adherence to appointment keeping, the pre-post intervention change was worse among ASG members (a decline of 67% to 45%) than among control youth (a decline of 55% to 47%), and this difference was statistically significant in both the crude and minimally adjusted models. When we assessed the sensitivity of our results to the number of control youth selected per ASG member from each matching stratum, our sample size of control youth decreased to 136 when we selected up to one control youth per ASG member and our sample size increased to 244 when we selected up to three control youths per ASG member, but our findings did not substantively change.

## Discussion

Our retrospective cohort analysis of a youth-focused combined peer-support and economic incentive program did not identify any association between Adolescent Support Group membership and improved clinical outcomes. Two previous small pilot studies have assessed similar interventions, with one observing no clinical improvements and the other not reporting on clinical outcomes [21, 22]. However, both studies identified potential non-clinical benefits to program participation, including psychosocial improvements and financial benefits, through qualitative interviews.

In our previous assessment on the implementation of the ASG program, we noted that a number of adaptations and infidelities to the program occurred over time [25]. Because recruitment into the ASG program was left to the nurses' discretion, we observed a much wider range of ages and a much wider range of groups sizes than expected. This variability in implementation contributed to key challenges in the current evaluation. For example–because a large proportion of the ASG members fell outside the target age range of 15–25, we retained youth born between 1987 and 2012 in our analytic sample. This approach avoided excluding an excessively large proportion of ASG members, but also meant that a large proportion of youth in both the intervention and control groups fell outside of the 15–25 target age range.

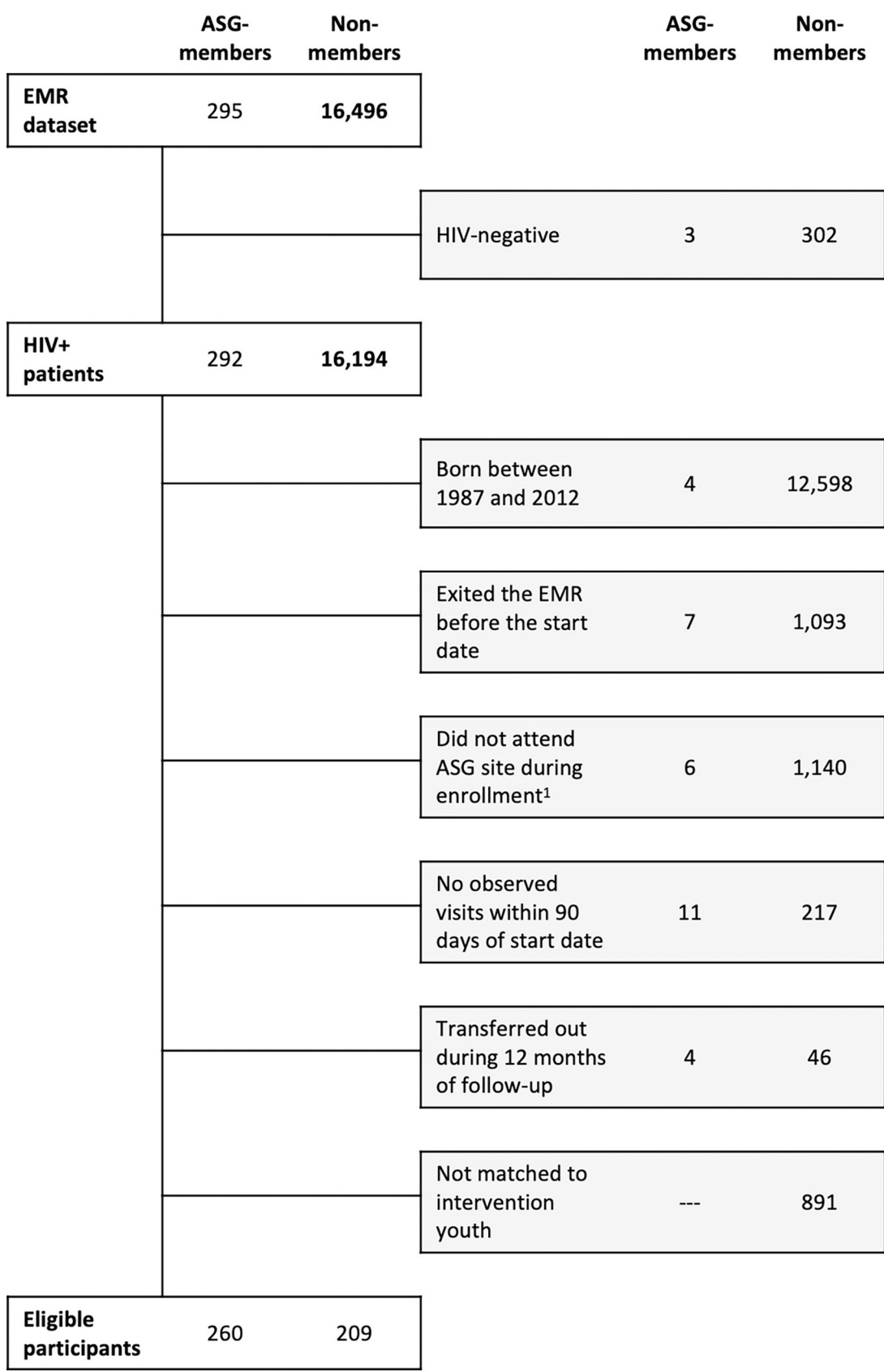

 

**Fig 1. Flowchart of study participants.** [1]ASG members were excluded if they did not attend their assigned site during the enrollment period. Non-members were excluded if they did not attend any participating sites during the enrollment period.

**Table 2. Demographic characteristics of ASG members and control youth (N = 469).**

|  | ASG member | Controls | p-value |
|---|---|---|---|
|  | N = 260 | N = 209 |  |
| District |  |  | 0.219 |
| Kayonza | 88 (33.8%) | 87 (41.6%) |  |
| Kirehe | 74 (28.5%) | 51 (24.4%) |  |
| Burera | 98 (37.7%) | 71 (34.0%) |  |
| Sex |  |  | 0.053 |
| Female | 145 (55.8%) | 135 (64.6^) |  |
| Male | 115 (44.2%) | 74 (35.4%) |  |
| Age[1] |  |  | <0.001 |
| <15 | 39 (15.0%) | 52 (24.9%) |  |
| 15–19 | 109 (41.9%) | 60 (28.7%) |  |
| 20–25 | 100 (38.5%) | 74 (35.4%) |  |
| >25 | 12 (4.6%) | 23 (11.0%) |  |
| Patent type |  |  | 0.350 |
| Pediatric | 208 (80.0%) | 156 (74.6%) |  |
| PMTCT mother | 10 (3.8%) | 12 (5.7%) |  |
| Other | 42 (16.2%) | 41 (19.6%) |  |
| Distance to health center |  |  | 0.020 |
| <2 km | 81 (31.2%) | 48 (23.0%) |  |
| 2-5km | 118 (45.4%) | 86 (41.1%) |  |
| ≥5km | 36 (13.8%) | 48 (23.0%) |  |
| Missing | 25 (9.6%) | 27 (12.9%) |  |
| Duration in HIV program[1,2] |  |  | 0.041 |
| <1 year | 11 (4.2%) | 12 (5.7%) |  |
| 1–5 years | 51 (19.6%) | 60 (28.7%) |  |
| ≥5 years | 198 (76.2%) | 137 (65.6%) |  |
| ART treatment category[1] |  |  | 0.659 |
| No initiation recorded | 1 (0.4%) | 2 (1.0%) |  |
| First line regimen | 235 (90.4%) | 186 (89.0%) |  |
| Second Line Regimen | 21 (8.1%) | 16 (7.7%) |  |
| Unknown Regimen | 3 (1.2%) | 5 (2.4%) |  |
| Last viral load results[1] |  |  | 0.115 |
| <20 | 118 (45.4%) | 103 (49.3%) |  |
| ≥20 to <1000 | 52 (20.0%) | 40 (19.1%) |  |
| ≥1000 | 48 (18.5%) | 23 (11.0%) |  |
| No VL test within two years | 42 (16.2%) | 43 (20.6%) |  |
| BMI[1] |  |  | 0.248 |
| Underweight (<18.5) | 59 (22.7%) | 62 (29.7%) |  |
| Normal weight (18.5-<25) | 133 (51.2%) | 92 (44.0%) |  |
| Overweight (≥25) | 41 (15.8%) | 29 (13.9%) |  |
| Missing | 27 (10.4%) | 26 (12.4%) |  |

[1]Assessed at the start of follow-up.

[2]Duration in HIV program refers to time since the youth's first recorded visit to receive clinical care for HIV.

 

**Table 3. Twelve-month clinical outcomes (N = 469).**

|  | ASG members N = 260 | | Control Youth N = 209 | |
|---|---|---|---|---|
|  | N | % | N | % |
| Death-free retention[1] | 242 | 93% | 196 | 94% |
| Death-free retention with active follow-up[2] | 206 | 79% | 163 | 78% |
| ≥80% adherence to appointment keeping | 109 | 42% | 90 | 43% |
| Viral load suppression (<20 copies/mL)[3] | 98 | 48% | 86 | 51% |

[1] Patients were considered to be retained if they had at least one observed clinical visit in the EMR after the twelve month follow up, if ≤90 days had passed since their final appointment, or, if a final appointment had not been scheduled after their last observed visit, if ≤210 days had passed from their final observed visit.

[2] Patients were under active follow-up at the end of the study period if they were observed a) <90 days from their last appointment scheduled during the study period or b) <210 days from last observed visit occurring during the study period or b) >210 days had passed from final observed visit. Patients with substantial treatment gaps during the study period could be retained without permanent loss to follow-up, but not meet the criteria for active follow-up.

[3] 56 intervention youth and 39 control youth did not have viral load test results within 6–18 months of the enrollment date and were excluded from the complete case analysis.

Similarly, in some facilities nurses appeared to recruit almost all youth living with HIV to the ASG program, leaving few youth available to be controls from that sites and preventing us from matching each ASG member with an equal number of control youth. In addition to creating methodologic complications, these infidelities to program implementation could have also made the intervention less successful and partially explain our null findings. Finally, in our previous work where we used programmatic data to assess group-level performance on key indicators, nurses reported much higher group achievement than what was observed in the EMR. For example, based on programmatic data, over half of the ASG groups reported 100% viral suppressions rates in the first year of the program [25]. There are three likely explanations for this. First, nurses may misreported group performance because it was difficult and time consuming to calculate and track indicators, particularly those around viral suppression. Second, nurses could have over-reported group performance in an effort to help the youth at their facility access more financial resources. These first two explanations point to implementation issues that could have negatively impacted the program by distorting the link between group behavior and money earned, therefore reducing the motivational power of the incentives. Small adjustments to the award scheme may have made the program easier to implement, better understood among youth, and more in line with behavioral economics theory

**Table 4. Results from generalized mixed models with a logit link and random intercepts for health facility (N = 469).**

|  | Crude analysis | | Minimally Adjusted Analysis[1] | | Fully Adjusted Analysis[2] | |
|---|---|---|---|---|---|---|
|  | OR (95% CI) | p-value | OR (95% CI) | p-value | OR (95% CI) | p-value |
| Death-free retention | 1.04 (0.47, 2.34) | 0.917 | 1.05 (0.43, 2.57) | 0.917 | 1.15 (0.46, 2.86) | 0.764 |
| Death-free retention under active follow-up | 1.18 (0.71, 1.97) | 0.517 | 1.41 (0.80, 2.49) | 0.240 | 1.45 (0.82, 2.57) | 0.205 |
| ≥80% adherence to appointment keeping | 0.91 (0.58, 1.42) | 0.672 | 0.98 (0.60, 1.58) | 0.925 | 0.98 (0.60, 1.58) | 0.923 |
| Viral load suppression (<20 copies/mL), complete case (N = 374) | 0.85 (0.54, 1.34) | 0.487 | 1.09 (0.64, 1.87) | 0.745 | 1.13 (0.66, 1.94) | 0.652 |
| Viral load suppression (<20 copies/mL), multiply imputed | 0.91 (0.60, 1.39) | 0.666 | 1.13 (0.68, 1.88) | 0.626 | 1.11 (0.67, 1.83) | 0.677 |

[1] Minimally adjusted analysis includes all variables with p<0.20 in Table 1 (sex, age, distance from health facility, having been on treatment for over 5 years, results of previous viral load test).

[2] Fully adjusted analysis includes all variables in Table 1 except ART treatment category at start of ASG due to positivity issues.

**Table 5. Difference-in-difference estimates from generalized mixed models with a logit link and random intercepts for health facility and patient (N = 413).**

| | ASG members N = 231 | | | | Control Youth N = 182 | | | | Coefficients for interaction term | | | |
|---|---|---|---|---|---|---|---|---|---|---|---|---|
| | Pre | | Post | | Pre | | Post | | Crude analysis | | Minimally Adjusted Analysis | |
| | N | % | N | % | N | % | N | % | β (95% CI) | p-value | β (95% CI) | p-value |
| Death-free retention under active follow-up | 215 | 93% | 190 | 82% | 166 | 91% | 146 | 80% | -0.16 (-1.12, 0.79) | 0.738 | -0.15 (-1.13, 0.83) | 0.763 |
| ≥80% adherence to appointment keeping[1] | 155 | 67% | 103 | 45% | 99 | 55% | 86 | 47% | -0.92 (-1.61, -0.23) | 0.009 | -0.91 (-1.60, -0.22) | 0.010 |
| Viral load suppression (<20 copies/mL), complete case (N = 383[2]) | 97 | 60% | 92 | 50% | 78 | 58% | 79 | 53% | -0.39 (-1.23, 0.46) | 0.368 | -0.37 (1.13, 0.39) | 0.340 |
| Viral load suppression (<20 copies/mL), multiply imputed | — | 57% | — | 51% | — | 56% | — | 53% | -0.21 (-0.98, 0.56) | 0.590 | -0.22 (-0.96, 0.51) | 0.552 |

[1]One control youth did not have any recorded scheduled visit dates during the pre-period.

[2] Includes 135 control youth and 161 intervention youth in the pre-period and 148 control youth and 183 intervention youth in the post-period.

[27]. For example, rather than incentivizing attendance through the stepwise reward system, promising the group a fixed amount of money each quarter and then implementing penalties for missed appointments might allow the youth to better understand the "cost" of missing appointments. Similarly, providing awards for suppressed viral loads as soon as test results became available would have been both easier to implement and resulted in timelier rewards for good behavior. Finally, incomplete record keeping in the EMR could led us to underestimate attendance and viral suppression; however, we would not expect the record keeping among ASG members to be worse than among controls, and so this hypothesis alone cannot be used to explain our null findings.

When comparing our study population to a recent national survey, the prevalence of viral load suppression, defined as <1,000 copies/mL, was similar to that of other youth aged 15–24 living with HIV (60.6%) but much lower to that of youth who were on ART (86.0%) [28]. This comparison is surprising since all but three of our study participants had initiated ART at the start of the study. Our population's poor clinical outcomes relative to the national average could be partially explained by the large proportion of our study population who are male or reside in Eastern Province, both of which are documented risk factors for elevated viral loads [28]. However, it may also be explained by the large proportion of pediatric patients in our study, most of whom had presumably been perinatally infected. There is some evidence from the United States that behaviorally and perinatally infected youth face distinct challenges to adherence and different patterns of viral suppression [29–31]. Less is known how behaviorally and perinatally infected youth may differ in LMICs. However, in Rwanda, as in many other African countries, a substantial proportion of behaviorally infected youth are young women who are identified and supported through the PMTCT program. In contrast, perinatally infected youth in this setting often face complex sets of socioeconomic challenges related to orphanhood and subsequent poverty and homelessness [32]. Similarly, perinatally infected youth in LMICs often face particularly challenging transitions from pediatric to adult HIV programs [33], and it is possible that the larger reduction in appointment attendance among ASG members compared to non-members observed in our difference-in-differences analysis could have resulted from nurses preferentially recommending youth who were actively undergoing this difficult process into the ASG program. Even though our intervention did not produce the desired clinical impact, continued efforts to provide clinical and economic support for these youth with HIV are warranted.

This study did suffer from several limitations. First, retrospective identification of beneficiaries was challenging, and only 86% of participants could be matched to the EMR with strong confidence. In addition to reducing power, poor matching could have also resulted in the misclassification of some youth to the intervention or control groups, which we would expect to bias our results to the null. Second, EMR data is vulnerable to missing and misrecorded data. Although this limitation is most obvious for our outcome of viral load suppression, which was missing for 20% of youth, incorrect or missing information about visit dates would have also biased our estimates of the active follow-up and visits adherence downward. Similarly, it is likely that the three youth without data on ART regimen likely had initiated treatment but that this information was not properly recorded in the EMR. Third, it was challenging to construct a control group that was similar to the intervention group. Ultimately these two groups differed significantly in terms of age, distance to the health facility, and duration in the HIV treatment program and likely also differed in terms of unmeasured covariates, especially socioeconomic status. Finally, this analysis only considered clinical outcomes and did not assess economic or psychosocial benefits. To complement this retrospective cohort, we have conducted qualitative interviews with youth beneficiaries to understand youth perspectives on the program, which we hope will provide insights on why the intervention did not produce the desired clinical impact, what non-clinical benefits may result from participation, and allow us to improve future iterations of this program. Future programs should consider pairing an improved intervention with a more rigorous, prospective evaluation that includes an appropriate control group and assesses relevant non-clinical outcomes.

## Conclusions

The Adolescent Support Group program did not improve retention, appointment adherence, or viral suppression among HIV positive youth in rural Rwanda. However, this population remains vulnerable to poor clinical outcomes. Additional research is needed to understand the contextual clinical and non-clinical challenges faced by these youth and to identify opportunities to address these challenges. Future complex interventions to address the biomedical, psychosocial, and socioeconomic challenges faced by these youth should incorporate theory from implementation science and behavioral economics as well as youth perspectives in their design.

## Supporting information

**S1 Checklist. Inclusivity in global research.**
(DOCX)

## Acknowledgments

The authors gratefully acknowledge the support of Partners In Health/Inshuti Mu Buzima for this work. Our acknowledgment also goes to ASG nurses and respective health facilities for support during this study.

## Author Contributions

**Conceptualization:** Dale A. Barnhart, Jean d'Amour Ndahimana, Vincent K. Cubaka, Bethany Hedt-Gauthier.

**Data curation:** Dale A. Barnhart, Josée Uwamariya, Gerardine Mukesharurema, Todd Anderson.

**Formal analysis:** Dale A. Barnhart.

**Funding acquisition:** Dale A. Barnhart, Jean d'Amour Ndahimana.

**Methodology:** Dale A. Barnhart, Bethany Hedt-Gauthier.

**Project administration:** Josée Uwamariya.

**Supervision:** Dale A. Barnhart, Jean d'Amour Ndahimana, Vincent K. Cubaka, Bethany Hedt-Gauthier.

**Validation:** Jean Népomuscène Nshimyumuremyi, Gerardine Mukesharurema.

**Writing – original draft:** Dale A. Barnhart.

**Writing – review & editing:** Dale A. Barnhart, Josée Uwamariya, Jean Népomuscène Nshimyumuremyi, Gerardine Mukesharurema, Todd Anderson, Jean d'Amour Ndahimana, Vincent K. Cubaka, Bethany Hedt-Gauthier.

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
