## [Decision Letter · Decision Letter 0]

2 Feb 2022

PGPH-D-21-01154

Receipt of a combined economic and peer support intervention and clinical outcomes among HIV-positive youth in rural Rwanda: A retrospective cohort

Dear Dr. Barnhart,

Thank you for submitting your manuscript to PLOS Global Public Health. After careful consideration, we feel that it has merit but does not fully meet PLOS Global Public Health’s publication criteria as it currently stands. Therefore, we invite you to submit a revised version of the manuscript that addresses the points raised during the review process.

We look forward to receiving your revised manuscript.

Kind regards,

Kévin Jean

Academic Editor

Journal Requirements:

2. Please update the completed 'Competing Interests' statement, including any COIs declared by your co-authors. If you have no competing interests to declare, please state "The authors have declared that no competing interests exist". Otherwise please declare all competing interests beginning with the statement "I have read the journal's policy and the authors of this manuscript have the following competing interests:"

Additional Editor Comments (if provided):

Reviewers' comments:

Reviewer's Responses to Questions

**Comments to the Author**

1. Does this manuscript meet PLOS Global Public Health’s publication criteria? Is the manuscript technically sound, and do the data support the conclusions? The manuscript must describe methodologically and ethically rigorous research with conclusions that are appropriately drawn based on the data presented.

Reviewer #1: Yes

Reviewer #2: Partly

2. Has the statistical analysis been performed appropriately and rigorously?

Reviewer #1: Yes

Reviewer #2: Yes

3. Have the authors made all data underlying the findings in their manuscript fully available (please refer to the Data Availability Statement at the start of the manuscript PDF file)?

Reviewer #1: No

Reviewer #2: Yes

4. Is the manuscript presented in an intelligible fashion and written in standard English?

Reviewer #1: Yes

Reviewer #2: No

5. Review Comments to the Author

Reviewer #1: Thank you for presenting important findings that showed no effect of an intervention on important clinical outcomes. We need to see more evaluations of public health interventions that show no effect so that we can adapt and refine our interventions.

The paper is concise and well written.

The analysis reported in this paper is as rigorous as it can be using the routine data methods employed and this could result in many biases which you rightly discuss in your limitations. I am happy that although not the "perfect" analysis it is as robust as it can be in the circumstances.

My major concern with the paper is the inability to really learn from the negative findings as the assessment of the intervention is not fully described in this paper. From what is reported this is the subject of another paper, which is referenced but as submitted rather than published, and so the reader cannot access that paper to understand the issues. I would request that a more systematic description of the implementation be included within this paper to make it much stronger. For example were the failures in implementation of the programme as designed uniform across all facilities or were there differences that could be further explored?

The intervention is also quite complex and I feel that a table showing the intended design of the intervention and the actual implementation would be helpful to readers and may allow for the differences in fidelity to be better quantified. This would make it easier to understand which aspects were delivered as expected and which not- this is unclear in the current text.

Reviewer #2: The manuscript by Barnhart and colleagues aims at presenting programmatic outcomes, in Rwanda, linked to retention and to virological success in adolescents who benefited from a dedicated support programme (the Adolescent Support Group (ASG) programme), and adolescents who did not benefit from this dedicated support programme.

Briefly, the ASG programme was initiated in 2017 in one region and in 2018 in the two other regions considered. “Economically vulnerable” youth, aged 15 to 24 years of age, were addressed by nurses to the ASG. Each ASG consisted of 10 to 15 members and met once a month to receive peer-support. A key aspect of the ASG was the financial motivation; incentives were deposited in a saving account in case of specific target attainment (i.e. quarterly pharmacy attendance, viral suppression). Then, each member of the ASG received a proportion of the money available on the saving account, depending on the performances of the ASG (100% achievement translated into obtaining 100% of the savings, 90% to 100% achievements translated into obtaining 80% of the savings…).

The strategy to offer group-incentives to motivate retention in care is very seducing and worth investigating. However, the article is very unclear and confusing.

Major comments

The study is based on cases and matched controls. However, the choice of the cases is unclear in terms of calendar period and also in terms of timing in the HIV infection. One would expect that controls are the adolescents who were enrolled in the ASG from one date to another date. But there is no such statement in the manuscript. One would then expect the controls to be adolescents not enrolled in the ASG during the same calendar period. But again, the enrolment period is never mentioned.

The lack of information regarding the enrolment period is even more surprising as the definition of outcomes are time-dependent; Indeed, the first two outcomes of interest were death-free retention at 12 months, and death-free retention with active follow-up at 12 months. It is unclear what “12 months” refers to in this study. Is it 12 months after enrolment in ASG, after enrolment in the present study, after ART initiation… The same way “active follow-up” is defined as follows: “Patients were under active follow-up at the end of the study period if they were observed <90 days from their last appointment scheduled during the study period or, if a final appointment had not been scheduled after their last observed visit during the study period, <210 days from last observed visit occurring during the study period”. This sentence is very hard to understand, and again refers to the “study period” which is never defined. The section from lines 161 to 176 must be reviewed.

The authors mention that controls were matched by facility, year of birth and paediatric infection status. The controls should also be matched by calendar period. Not mentioning this suggests that that participants could be compared over calendar periods that are not similar, and worryingly that the considered calendar period for controls can be before 2017 when ASG were implemented. If this is the case, not considering strictly the same calendar period for cases and controls is a real issue as care conditions were not comparable.

In Table 1, it appears that 3 adolescents are not yet on ART. The ASG measures rely on pharmacy visits and viral load assessment. I do not understand how those not on ART can fit in the ASG. This is surprising, and I would recommend removing these 3 individuals.

In the section entitled The adolescent Support Group program, it is mentioned that implementation started in 2017 and 2018 depending on the region, and that adolescents can be enrolled in ASG if aged 15 to 25 years (line 79-81). In the Eligibility section (line 126), it is mentioned that patients born between 1987 and 2012 were eligible. The enrolment period is unspecified, but if we assume the enrolment took place in 2018, eligibility criterion makes that the patients could then be as young as 6 years of age and as old as 31 years. This does not suit the target years 15 to 24. Again, this is very confusing and things must be clarified.

Also, what are the reasons for adolescents not to be enrolled in the ASG? Did some decline? Were some not invited? If these two situations exist, it could be useful to distinguish these 2 groups of controls as they may not present the same characteristics.

It would also be helpful to present group characteristics, in terms of pharmacy attendance and virological success, and in terms of payment received. Were there some ASG with extremely good outcomes and some with poor outcomes? Group performance level could be considered as a potential cofactor.

In the method section, the authors mention “multiple imputation” procedures when dealing with viral suppression outcomes. I am not very comfortable with the fact that outcome are imputed, based on available demographic and clinical characteristics; and then the association between these same demographic and clinical factors and the imputed outcome is investigated. In table 1, the authors state that some demographic characteristics were also missing. Did the authors perform multiple imputations on the demographic characteristics also? This is not clear and this would be wiser to impute these data rather than the outcomes.

In the method section, I do not understand the difference-in-difference analysis. I really do not understand what the authors did.

Minor comments

Although it seems the case, it would be useful, for a better understanding of the ASG programme, to know if groups were of mixed gender and mixed age as well.

To make things easier to read, the description of payment to the members of the ASG, based on the group performances, could be presented in a table.

In table 1, “duration in HIV program” is not explicit. Please define what this duration really defines.

In Table 1, “last viral load result” would make more sense if a delay was also considered (e.g. measured within 12 months from enrolment).

The conclusion of the manuscript seems to be that ASG did not prove useful in terms of retention or virological success. However, the manuscript lacks of clear information, definition and data to support that conclusion. First, the selection procedures and enrolment criteria must be clearly stated, especially in terms of calendar period. Then, I would recommend presenting some ASG outcomes by group and not just by individuals.

6. PLOS authors have the option to publish the peer review history of their article (what does this mean?). If published, this will include your full peer review and any attached files.

**Do you want your identity to be public for this peer review?** For information about this choice, including consent withdrawal, please see our Privacy Policy.

Reviewer #1: No

Reviewer #2: No

---

## [Decision Letter · Decision Letter 1]

16 May 2022

Receipt of a combined economic and peer support intervention and clinical outcomes among HIV-positive youth in rural Rwanda: A retrospective cohort

PGPH-D-21-01154R1

Dear Dr Barnhart,

We are pleased to inform you that your manuscript 'Receipt of a combined economic and peer support intervention and clinical outcomes among HIV-positive youth in rural Rwanda: A retrospective cohort' has been provisionally accepted for publication in PLOS Global Public Health.

Best regards,

Kévin Jean

Academic Editor

Reviewer Comments (if any, and for reference):

Reviewer's Responses to Questions

**Comments to the Author**

1. If the authors have adequately addressed your comments raised in a previous round of review and you feel that this manuscript is now acceptable for publication, you may indicate that here to bypass the “Comments to the Author” section, enter your conflict of interest statement in the “Confidential to Editor” section, and submit your "Accept" recommendation.

Reviewer #2: All comments have been addressed

2. Does this manuscript meet PLOS Global Public Health’s publication criteria? Is the manuscript technically sound, and do the data support the conclusions? The manuscript must describe methodologically and ethically rigorous research with conclusions that are appropriately drawn based on the data presented.

Reviewer #2: Yes

3. Has the statistical analysis been performed appropriately and rigorously?

Reviewer #2: Yes

4. Have the authors made all data underlying the findings in their manuscript fully available (please refer to the Data Availability Statement at the start of the manuscript PDF file)?

Reviewer #2: Yes

5. Is the manuscript presented in an intelligible fashion and written in standard English?

Reviewer #2: Yes

6. Review Comments to the Author

Reviewer #2: The authors have clearly adressed all the points raised during the review of their manuscript.

The manuscript is now easier to read and it clearly exposes the shift between what was planned and what really happened in the field.

It provides interesting data and I recommend it for publication.

7. PLOS authors have the option to publish the peer review history of their article (what does this mean?). If published, this will include your full peer review and any attached files.

**Do you want your identity to be public for this peer review?** For information about this choice, including consent withdrawal, please see our Privacy Policy.

Reviewer #2: No
